# Low-Rank Matrix and Tensor Completion via Adaptive Sampling

**Akshay Krishnamurthy**
Computer Science Department
Carnegie Mellon University
Pittsburgh, PA 15213
akshaykr@cs.cmu.edu

**Aarti Singh**
Machine Learning Department
Carnegie Mellon University
Pittsburgh, PA 15213
aartisingh@cs.cmu.edu

## Abstract

We study low rank matrix and tensor completion and propose novel algorithms that employ adaptive sampling schemes to obtain strong performance guarantees. Our algorithms exploit adaptivity to identify entries that are highly informative for learning the column space of the matrix (tensor) and consequently, our results hold even when the row space is highly coherent, in contrast with previous analyses. In the absence of noise, we show that one can exactly recover a $n \times n$ matrix of rank $r$ from merely $\Omega(nr^{3/2}\log(r))$ matrix entries. We also show that one can recover an order $T$ tensor using $\Omega(nr^{T-1/2}T^2\log(r))$ entries. For noisy recovery, our algorithm consistently estimates a low rank matrix corrupted with noise using $\Omega(nr^{3/2}\text{polylog}(n))$ entries. We complement our study with simulations that verify our theory and demonstrate the scalability of our algorithms.

## 1 Introduction

Recently, the machine learning and signal processing communities have focused considerable attention toward understanding the benefits of adaptive sensing. This theme is particularly relevant to modern data analysis, where adaptive sensing has emerged as an efficient alternative to obtaining and processing the large data sets associated with scientific investigation. These empirical observations have lead to a number of theoretical studies characterizing the performance gains offered by adaptive sensing over conventional, passive approaches. In this work, we continue in that direction and study the role of adaptive data acquisition in low rank matrix and tensor completion problems.

Our study is motivated not only by prior theoretical results in favor of adaptive sensing but also by several applications where adaptive sensing is feasible. In recommender systems, obtaining a measurement amounts to asking a user about an item, an interaction that has been deployed in production systems. Another application pertains to network tomography, where a network operator is interested in inferring latencies between hosts in a communication network while injecting few packets into the network. The operator, being in control of the network, can adaptively sample the matrix of pair-wise latencies, potentially reducing the total number of measurements. In particular, the operator can obtain full columns of the matrix by measuring from one host to all others, a sampling strategy we will exploit in this paper.

Yet another example centers around gene expression analysis, where the object of interest is a matrix of expression levels for various genes across a number of conditions. There are typically two types of measurements: low-throughput assays provide highly reliable measurements of single entries in this matrix while high-throughput microarrays provide expression levels of all genes of interest across operating conditions, thus revealing entire columns. The completion problem can be seen as a strategy for learning the expression matrix from both low- and high-throughput data while minimizing the total measurement cost.

## 1.1 Contributions

We develop algorithms with theoretical guarantees for three low-rank completion problems. The algorithms find a small subset of columns of the matrix (tensor) that can be used to reconstruct or approximate the matrix (tensor). We exploit adaptivity to focus on highly informative columns, and this enables us to do away with the usual incoherence assumptions on the row-space while achieving competitive (or in some cases better) sample complexity bounds. Specifically our results are:

1. In the absence of noise, we develop a streaming algorithm that enjoys both low sample requirements and computational overhead. In the matrix case, we show that $\Omega(nr^{3/2}\log r)$ adaptively chosen samples are sufficient for exact recovery, improving on the best known bound of $\Omega(nr^2\log^2 n)$ in the passive setting [21]. This also gives the first guarantee for matrix completion with coherent row space.

2. In the tensor case, we establish that $\Omega(nr^{T-1/2}T^2\log r)$ adaptively chosen samples are sufficient for recovering a $n \times \ldots \times n$ order $T$ tensor of rank $r$. We complement this with a necessary condition for tensor completion under random sampling, showing that our adaptive strategy is competitive with *any* passive algorithm. These are the first sample complexity upper and lower bounds for exact tensor completion.

3. In the noisy matrix completion setting, we modify the adaptive column subset selection algorithm of Deshpande *et al.* [10] to give an algorithm that finds a rank-$r$ approximation to a matrix using $\Omega(nr^{3/2}\text{polylog}(n))$ samples. As before, the algorithm does not require an incoherent row space but we are no longer able to process the matrix sequentially.

4. Along the way, we improve on existing results for subspace detection from missing data, the problem of testing if a partially observed vector lies in a known subspace.

## 2 Related Work

The matrix completion problem has received considerable attention in recent years. A series of papers [6, 7, 13, 21], culminating in Recht's elegant analysis of the nuclear norm minimization program, address the exact matrix completion problem through the framework of convex optimization, establishing that $\Omega((n_1 + n_2)r\max\{\mu_0, \mu_1^2\}\log^2(n_2))$ randomly drawn samples are sufficient to exactly identify an $n_1 \times n_2$ matrix with rank $r$. Here $\mu_0$ and $\mu_1$ are parameters characterizing the *incoherence* of the row and column spaces of the matrix, which we will define shortly. Candes and Tao [7] proved that under random sampling $\Omega(n_1 r\mu_0 \log(n_2))$ samples are necessary, showing that nuclear norm minimization is near-optimal.

The noisy matrix completion problem has also received considerable attention [5, 17, 20]. The majority of these results also involve some parameter that quantifies how much information a single observation reveals, in the same vein as incoherence.

Tensor completion, a natural generalization of matrix completion, is less studied. One challenge stems from the NP-hardness of computing most tensor decompositions, pushing researchers to study alternative structure-inducing norms in lieu of the nuclear norm [11, 22]. Both papers derive algorithms for tensor completion, but neither provide sample complexity bounds for the noiseless case.

Our approach involves adaptive data acquisition, and consequently our work is closely related to a number of papers focusing on using adaptive measurements to estimate a sparse vector [9, 15]. In these problems, specifically, problems where the sparsity basis is known a priori, we have a reasonable understanding of how adaptive sampling can lead to performance improvements. As a low rank matrix is sparse in its unknown eigenbasis, the completion problem is coupled with learning this basis, which poses a new challenge for adaptive sampling procedures.

Another relevant line of work stems from the *matrix approximations* literature. Broadly speaking, this research is concerned with efficiently computing a structured matrix, i.e. sparse or low rank, that serves as a good approximation to a fully observed input matrix. Two methods that apply to the missing data setting are the Nystrom method [12, 18] and entrywise subsampling [1]. While the sample complexity bounds match ours, the analysis for the Nystrom method has focused on positive-semidefinite kernel matrices and requires incoherence of both the row and column spaces. On the other hand, entrywise subsampling is applicable, but the guarantees are weaker than ours.

It is also worth briefly mentioning the vast body of literature on column subset selection, the task of approximating a matrix by projecting it onto a few of its columns. While the best algorithms, namely volume sampling [14] and sampling according to statistical leverages [3], do not seem to be readily applicable to the missing data setting, some algorithms are. Indeed our procedure for noisy matrix completion is an adaptation of an existing column subset selection procedure [10].

Our techniques are also closely related to ideas employed for subspace detection – testing whether a vector lies in a known subspace – and subspace tracking – learning a time-evolving low-dimensional subspace from vectors lying close to that subspace. Balzano *et al.* [2] prove guarantees for subspace detection with known subspace and a partially observed vector, and we will improve on their result en route to establishing our guarantees. Subspace tracking from partial information has also been studied [16], but little is known theoretically about this problem.

## 3 Definitions and Preliminaries

Before presenting our algorithms, we clarify some notation and definitions. Let $M \in \mathbb{R}^{n_1 \times n_2}$ be a rank $r$ matrix with singular value decomposition $U\Sigma V^T$. Let $c_1, \ldots c_{n_2}$ denote the columns of $M$.

Let $\mathbb{M} \in \mathbb{R}^{n_1 \times \ldots \times n_T}$ denote an order $T$ tensor with canonical decomposition:

$$\mathbb{M} = \sum_{k=1}^{r} a_k^{(1)} \otimes a_k^{(2)} \otimes \ldots \otimes a_k^{(T)} \tag{1}$$

where $\otimes$ is the outer product. Define rank$(\mathbb{M})$ to be the smallest value of $r$ that establishes this equality. Note that the vectors $\{a_k^{(t)}\}_{k=1}^r$ need not be orthogonal, nor even linearly independent.

The mode-$t$ subtensors of $\mathbb{M}$, denoted $\mathbb{M}_i^{(t)}$, are order $T-1$ tensors obtained by fixing the $i$th coordinate of the $t$-th mode. For example, if $\mathbb{M}$ is an order 3 tensor, then $\mathbb{M}_i^{(3)}$ are the frontal slices.

We represent a $d$-dimensional subspace $U \subset \mathbb{R}^n$ as a set of orthonormal basis vectors $U = \{u_i\}_{i=1}^d$ and in some cases as $n \times d$ matrix whose columns are the basis vectors. The interpretation will be clear from context. Define the **orthogonal projection** onto $U$ as $\mathcal{P}_U v = U(U^T U)^{-1} U^T v$.

For a set $\Omega \subset [n]^1$, $c_\Omega \in \mathbb{R}^{|\Omega|}$ is the vector whose elements are $c_i, i \in \Omega$ indexed lexicographically. Similarly the matrix $U_\Omega \in \mathbb{R}^{|\Omega| \times d}$ has rows indexed by $\Omega$ lexicographically. Note that if $U$ is a orthobasis for a subspace, $U_\Omega$ is a $|\Omega| \times d$ matrix with columns $u_{i\Omega}$ where $u_i \in U$, rather than a set of orthonormal basis vectors. In particular, the matrix $U_\Omega$ need not have orthonormal columns.

These definitions extend to the tensor setting with slight modifications. We use the `vec` operation to unfold a tensor into a single vector and define the inner product $\langle x, y \rangle = \text{vec}(x)^T \text{vec}(y)$. For a subspace $U \subset \mathbb{R}^{\otimes n_i}$, we write it as a $(\prod n_i) \times d$ matrix whose columns are $\text{vec}(u_i)$, $u_i \in U$. We can then define projections and subsampling as we did in the vector case.

As in recent work on matrix completion [7, 21], we will require a certain amount of incoherence between the column space associated with $M$ ($\mathbb{M}$) and the standard basis.

**Definition 1.** *The **coherence** of an $r$-dimensional subspace $U \subset \mathbb{R}^n$ is:*

$$\mu(U) \triangleq \frac{n}{r} \max_{1 \leq j \leq n} ||\mathcal{P}_U e_j||^2 \tag{2}$$

*where $e_j$ denotes the $j$th standard basis element.*

In previous analyses of matrix completion, the *incoherence assumption* is that both the row and column spaces of the matrix have coherences upper bounded by $\mu_0$. When both spaces are incoherent, each entry of the matrix reveals roughly the same amount of information, so there is little to be gained from adaptive sampling, which typically involves looking for highly informative measurements. Thus the power of adaptivity for these problems should center around relaxing the incoherence assumption, which is the direction we take in this paper. Unfortunately, even under adaptive sampling, it is impossible to identify a rank one matrix that is zero in all but one entry without observing the entire matrix, implying that we cannot completely eliminate the assumption. Instead, we will retain incoherence on the column space, but remove the restrictions on the row space.

<div align="center">Algorithm 1: Sequential Tensor Completion $(\mathbb{M}, \{m_t\}_{t=1}^T)$</div>

---

1. Let $\mathcal{U} = \emptyset$.
2. Randomly draw entries $\Omega \subset \prod_{t=1}^{T-1}[n_t]$ uniformly with replacement w. p. $m_T / \prod_{t=1}^{T-1} n_t$.
3. For each mode-$T$ subtensor $\mathbb{M}_i^{(T)}$ of $\mathbb{M}$, $i \in [n_T]$:
   (a) If $||\mathbb{M}_{i\Omega}^{(T)} - \mathcal{P}_{\mathcal{U}_\Omega}\mathbb{M}_{i\Omega}^{(t)}||_2^2 > 0$:
      i. $\hat{\mathbb{M}}_i^{(T)} \leftarrow$ recurse on $(\mathbb{M}_i^{(T)}, \{m_t\}_{t=1}^{T-1})$
      ii. $\mathbb{U}_i \leftarrow \frac{\mathcal{P}_{\mathcal{U}^\perp}\hat{\mathbb{M}}_i^{(T)}}{||\mathcal{P}_{\mathcal{U}^\perp}\hat{\mathbb{M}}_i^{(T)}||}$. $\mathcal{U} \leftarrow \mathcal{U} \cup \mathbb{U}_i$.
   (b) Otherwise $\hat{\mathbb{M}}_i^{(T)} \leftarrow \mathcal{U}(\mathcal{U}_\Omega^T\mathcal{U}_\Omega)^{-1}\mathcal{U}_\Omega\mathbb{M}_{i\Omega}^{(T)}$
4. Return $\hat{\mathbb{M}}$ with mode-$T$ subtensors $\hat{\mathbb{M}}_i^{(T)}$.

---

## 4  Exact Completion Problems

In the matrix case, our sequential algorithm builds up the column space of the matrix by selecting a few columns to observe in their entirety. In particular, we maintain a candidate column space $\tilde{U}$ and test whether a column $c_i$ lives in $\tilde{U}$ or not, choosing to completely observe $c_i$ and add it to $\tilde{U}$ if it does not. Balzano *et al.* [2] observed that we can perform this test with a subsampled version of $c_i$, meaning that we can recover the column space using few samples. Once we know the column space, recovering the matrix, even from few observations, amounts to solving determined linear systems.

For tensors, the algorithm becomes recursive in nature. At the outer level of the recursion, the algorithm maintains a candidate subspace $\mathcal{U}$ for the mode $T$ subtensors $\mathbb{M}_i^{(T)}$. For each of these subtensors, we test whether $\mathbb{M}_i^{(T)}$ lives in $\mathcal{U}$ and recursively complete that subtensor if it does not. Once we complete the subtensor, we add it to $\mathcal{U}$ and proceed at the outer level. When the subtensor itself is just a column; we observe the columns in its entirety.

The pseudocode of the algorithm is given in Algorithm 1. Our first main result characterizes the performance of the tensor completion algorithm. We defer the proof to the appendix.

**Theorem 2.** *Let* $\mathbb{M} = \sum_{i=1}^r \otimes_{t=1}^T a_j^{(t)}$ *be a rank $r$ order-$T$ tensor with subspaces* $A^{(t)} = span(\{a_j^{(t)}\}_{j=1}^r)$. *Suppose that all of* $A^{(1)}, \dots A^{(T-1)}$ *have coherence bounded above by* $\mu_0$. *Set* $m_t = 36r^{t-1/2}\mu_0^{t-1}\log(2r/\delta)$ *for each $t$. Then with probability* $\geq 1 - 5\delta Tr^T$, *Algorithm 1 exactly recovers* $\mathbb{M}$ *and has expected sample complexity*

$$36(\sum_{t=1}^T n_t)r^{T-1/2}\mu_0^{T-1}\log(2r/\delta) \tag{3}$$

In the special case of a $n \times \dots \times n$ tensor of order $T$, the algorithm succeeds with high probability using $\Omega(nr^{T-1/2}\mu_0^{T-1}T^2\log(Tr/\delta))$ samples, exhibiting a linear dependence on the tensor dimensions. In comparison, the only guarantee we are aware of shows that $\Omega\left(\left(\prod_{t=2}^{T_1} n_t\right)r\right)$ samples are sufficient for consistent estimation of a noisy tensor, exhibiting a much worse dependence on tensor dimension [23]. In the noiseless scenario, one can unfold the tensor into a $n_1 \times \prod_{t=2}^T n_t$ matrix and apply any matrix completion algorithm. Unfortunately, without exploiting the additional tensor structure, this approach will scale with $\prod_{t=2}^T n_t$, which is similarly much worse than our guarantee. Note that the naïve procedure that does not perform the recursive step has sample complexity scaling with the product of the dimensions and is therefore much worse than the our algorithm.

The most obvious specialization of Theorem 2 is to the matrix completion problem:

**Corollary 3.** *Let* $M := U\Sigma V^T \in \mathbb{R}^{n_1 \times n_2}$ *have rank $r$, and fix $\delta > 0$. Assume $\mu(U) \leq \mu_0$. Setting* $m \triangleq m_2 \geq 36r^{3/2}\mu_0\log(\frac{2r}{\delta})$, *the sequential algorithm exactly recovers $M$ with probability at least* $1 - 4r\delta + \delta$ *while using in expectation*

$$36n_2r^{3/2}\mu_0\log(2r/\delta) + rn_1 \tag{4}$$

*observations. The algorithm runs in $O(n_1 n_2 r + r^3 m)$ time.*

A few comments are in order. Recht [21] guaranteed exact recovery for the nuclear norm minimization procedure as long as the number of observations exceeds $32(n_1+n_2)r\max\{\mu_0, \mu_1^2\}\beta\log^2(2n_2)$ where $\beta$ controls the probability of failure and $||UV^T||_\infty \le \mu_1\sqrt{r/(n_1 n_2)}$ with $\mu_1$ as another coherence parameter. Without additional assumptions, $\mu_1$ can be as large as $\mu_0\sqrt{r}$. In this case, our bound improves on his in its the dependence on $r$, $\mu_0$ and logarithmic terms.

The Nystrom method can also be applied to the matrix completion problem, albeit under non-uniform sampling. Given a PSD matrix, one uses a randomly sampled set of columns and the corresponding rows to approximate the remaining entries. Gittens showed that if one samples $O(r \log r)$ columns, then one can exactly reconstruct a rank $r$ matrix [12]. This result requires incoherence of both row and column spaces, so it is more restrictive than ours. Almost all previous results for exact matrix completion require incoherence of both row and column spaces.

The one exception is a recent paper by Chen *et al.* that we became aware of while preparing the final version of this work [8]. They show that sampling the matrix according to statistical leverages of the rows and columns can eliminate the need for incoherence assumptions. Specifically, when the matrix has incoherent column space, they show that by first estimating the leverages of the columns, sampling the matrix according to this distribution, and then solving the nuclear norm minimization program, one can recover the matrix with $\Omega(nr\mu_0\log^2 n)$ samples. Our result improves on theirs when $r$ is small compared to $n$, specifically when $\sqrt{r}\log r \le \log^2 n$, which is common.

Our algorithm is also very computationally efficient. Existing algorithms involve successive singular value decompositions ($O(n_1 n_2 r)$ per iteration), resulting in much worse running times.

The key ingredient in our proofs is a result pertaining to subspace detection, the task of testing if a subsampled vector lies in a subspace. This result, which improves over the results of Balzano *et al.* [2], is crucial in obtaining our sample complexity bounds, and may be of independent interest.

**Theorem 4.** *Let $U$ be a $d$-dimensional subspace of $\mathbb{R}^n$ and $y = x + v$ where $x \in U$ and $v \in U^\perp$. Fix $\delta > 0$, $m \ge \frac{8}{3}d\mu(U)\log\left(\frac{2d}{\delta}\right)$ and let $\Omega$ be an index set with entries sampled uniformly with replacement with probability $m/n$. Then with probability at least $1 - 4\delta$:*

$$\frac{m(1-\alpha) - d\mu(U)\frac{\beta}{(1-\gamma)}}{n}||v||_2^2 \le ||y_\Omega - \mathcal{P}_{U_\Omega}y_\Omega||_2^2 \le (1+\alpha)\frac{m}{n}||v||_2^2 \tag{5}$$

*Where $\alpha = \sqrt{2\frac{\mu(v)}{m}\log(1/\delta)} + 2\frac{\mu(v)}{3m}\log(1/\delta)$, $\beta = 6\log(d/\delta) + \frac{4}{3}\frac{d\mu(v)}{m}\log^2(d/\delta)$, $\gamma = \sqrt{\frac{8d\mu(U)}{3m}\log(2d/\delta)}$ and $\mu(v) = n||v||_\infty^2/||v||_2^2$.*

This theorem shows that if $m = \Omega(\max\{\mu(v), d\mu(U), d\sqrt{\mu(U)\mu(v)}\}\log d)$ then the orthogonal projection from missing data is within a constant factor of the fully observed one. In contrast, Balzano *et al.* [2] give a similar result that requires $m = \Omega(\max\{\mu(v)^2, d\mu(U), d\mu(U)\mu(v)\}\log d)$ to get a constant factor approximation. In the matrix case, this improved dependence on incoherence parameters brings our sample complexity down from $nr^2\mu_0^2\log r$ to $nr^{3/2}\mu_0\log r$. We conjecture that this theorem can be further improved to eliminate another $\sqrt{r}$ factor from our final bound.

## 4.1 Lower Bounds for Uniform Sampling

We adapt the proof strategy of Candes and Tao [7] to the tensor completion problem and establish the following lower bound for uniform sampling:

**Theorem 5** (Passive Lower Bound). *Fix $1 \le m, r \le \min_t n_t$ and $\mu_0 > 1$. Fix $0 < \delta < 1/2$ and suppose that we do not have the condition:*

$$-\log\left(1 - \frac{m}{\prod_{i=1}^T n_i}\right) \ge \frac{\mu_0^{T-1}r^{T-1}}{\prod_{i=2}^T n_i}\log\left(\frac{n_1}{2\delta}\right) \tag{6}$$

*Then there exist infinitely many pairs of distinct $n_1 \times \ldots \times n_T$ order-$T$ tensors $\mathbb{M} \ne \mathbb{M}'$ of rank $r$ with coherence parameter $\le \mu_0$ such that $\mathcal{P}_\Omega(\mathbb{M}) = \mathcal{P}_\Omega(\mathbb{M}')$ with probability at least $\delta$. Each entry is observed independently with probability $T = \frac{m}{\prod_{i=1}^T n_i}$.*

Theorem 5 implies that as long as the right hand side of Equation 6 is at most $\epsilon < 1$, and:

$$m \leq n_1 r^{T-1} \mu_0^{T-1} \log\left(\frac{n_1}{2\delta}\right)(1 - \epsilon/2) \tag{7}$$

then with probability at least $\delta$ there are infinitely many matrices that agree on the observed entries. This gives a necessary condition on the number of samples required for tensor completion. Note that when $T = 2$ we recover the known lower bound for matrix completion.

Theorem 5 gives a necessary condition under uniform sampling. Comparing with Theorem 2 shows that our procedure outperforms any passive procedure in its dependence on the tensor dimensions. However, our guarantee is suboptimal in its dependence on $r$. The extra factor of $\sqrt{r}$ would be eliminated by a further improvement to Theorem 5, which we conjecture is indeed possible.

For adaptive sampling, one can obtain a lower bound via a parameter counting argument. Observing the $(i_1, \ldots, i_T)$th entry leads to a polynomial equation of the form $\sum_k \prod_t a_k^{(t)}(i_t) = M_{i_1, \ldots, i_T}$. If $m < r(\sum_t n_t)$, this system is underdetermined showing that $\Omega((\sum_t n_t)r)$ observations are necessary for exact recovery, even under adaptive sampling. Thus, our algorithm enjoys sample complexity with optimal dependence on matrix dimensions.

## 5 Noisy Matrix Completion

Our algorithm for noisy matrix completion is an adaptation of the column subset selection (CSS) algorithm analyzed by Deshpande *et al.* [10]. The algorithm builds a candidate column space in rounds; at each round it samples additional columns with probability proportional to their projection on the orthogonal complement of the candidate column space.

To concretely describe the algorithm, suppose that at the beginning of the $l$th round we have a candidate subspace $U_l$. Then in the $l$th round, we draw $s$ additional columns according to the distribution where the probability of drawing the $i$th column is proportional to $||\mathcal{P}_{U_l^\perp} c_i||_2^2$. Observing these $s$ columns in full and then adding them to the subspace $U_l$ gives the candidate subspace $U_{l+1}$ for the next round. We initialize the algorithm with $U_1 = \emptyset$. After $L$ rounds, we approximate each column $c$ with $\hat{c} = U_L(U_{L\Omega}^T U_{L\Omega})^{-1} U_{L\Omega}^T c_\Omega$ and concatenate these estimates to form $\hat{M}$.

The challenge is that the algorithm cannot compute the sampling probabilities without observing entries of the matrix. However, our results show that with reliable estimates, which can be computed from few observations, the algorithm still performs well.

We assume that the matrix $M \in \mathbb{R}^{n_1 \times n_2}$ can be decomposed as a rank $r$ matrix $A$ and a random gaussian matrix $R$ whose entries are independently drawn from $\mathcal{N}(0, \sigma^2)$. We write $A = U\Sigma V^T$ and assume that $\mu(U) \leq \mu_0$. As before, the incoherence assumption is crucial in guaranteeing that one can estimate the column norms, and consequently sampling probabilities, from missing data.

**Theorem 6.** *Let $\Omega$ be the set of all observations over the course of the algorithm, let $U_L$ be the subspace obtained after $L = \log(n_1 n_2)$ rounds and $\hat{M}$ be the matrix whose columns $\hat{c}_i = U_L(U_{L\Omega}^T U_{L\Omega})^{-1} U_{L\Omega}^T c_{\Omega i}$. Then there are constants $c_1, c_2$ such that:*

$$||A - \hat{M}||_F^2 \leq \frac{c_1}{(n_1 n_2)}||A||_F^2 + c_2||R_\Omega||_F^2$$

*$\hat{M}$ can be computed from $\Omega((n_1 + n_2)r^{3/2}\mu(U) polylog(n_1 n_2))$ observations. In particular, if $||A||_F^2 = 1$ and $R_{ij} \sim \mathcal{N}(0, \sigma^2/(n_1 n_2))$, then there is a constant $c_\star$ for which:*

$$||A - \hat{A}||_F^2 \leq \frac{c_\star}{n_1 n_2}\left(1 + \sigma^2\left((n_1 + n_2)r^{3/2}\mu(U) polylog(n_1 n_2)\right)\right)$$

The main improvement in the result is in relaxing the assumptions on the underlying matrix $A$. Existing results for noisy matrix completion require that the energy of the matrix is well spread out across both the rows and the columns (i.e. incoherence), and the sample complexity guarantees deteriorate significantly without such an assumption [5, 17]. As a concrete example, Negahban and Wainwright [20] use a notion of spikiness, measured as $\sqrt{n_1 n_2}\frac{||A||_\infty}{||A||_F}$ which can be as large as $\sqrt{n_2}$ in our setup, e.g. when the matrix is zero except for on one column and constant across that column.

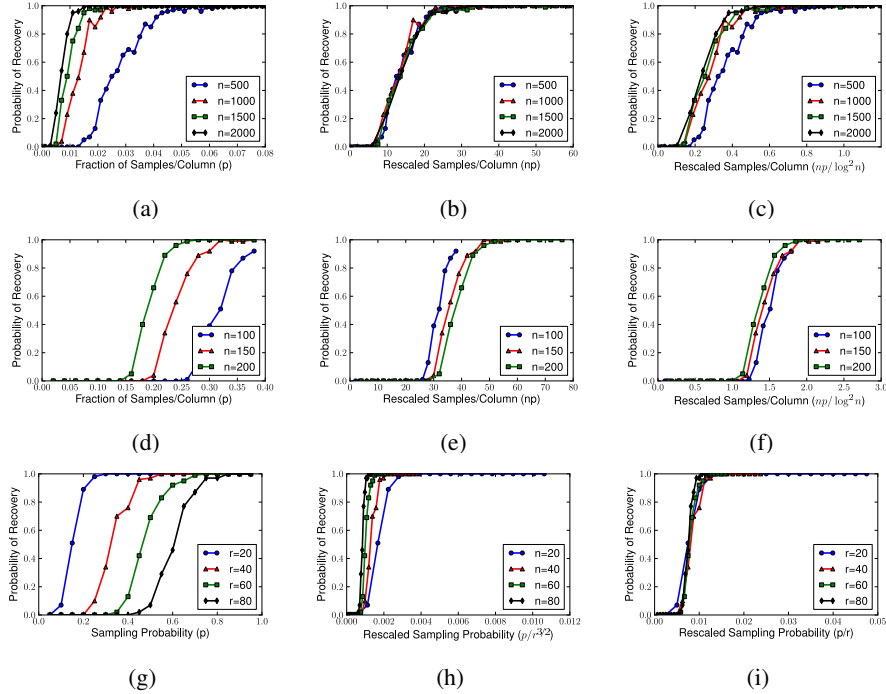

Figure 1: Probability of success curves for our noiseless matrix completion algorithm (top) and SVT (middle). Top: Success probability as a function of: Left: $p$, the fraction of samples per column, Center: $np$, total samples per column, and Right: $np\log^2 n$, expected samples per column for passive completion. Bottom: Success probability of our noiseless algorithm for different values of $r$ as a function of $p$, the fraction of samples per column (left), $p/r^{3/2}$ (middle) and $p/r$ (right).

The choices of $||A||_F^2 = 1$ and noise variance rescaled by $\frac{1}{n_1 n_2}$ enable us to compare our results with related work [20]. Thinking of $n_1 = n_2 = n$ and the incoherence parameter as a constant, our results imply consistent estimation as long as $\sigma^2 = \omega\left(\frac{n}{r^2 \text{polylog}(n)}\right)$. On the other hand, thinking of the spikiness parameter as a constant, [20] show that the error is bounded by $\frac{\sigma^2 nr \log n}{m}$ where $m$ is the total number of observations. Using the same number of samples as our procedure, their results implies consistency as long as $\sigma^2 = \omega(r\text{polylog}(n))$. For small $r$ (i.e. $r = O(1)$), our noise tolerance is much better, but their results apply even with fewer observations, while ours do not.

## 6  Simulations

We verify Corollary 3's linear dependence on $n$ in Figure 1, where we empirically compute the success probability of the algorithm for varying values of $n$ and $p = m/n$, the fraction of entries observed per column. Here we study square matrices of fixed rank $r = 5$ with $\mu(U) = 1$. Figure 1(a) shows that our algorithm can succeed with sampling a smaller and smaller fraction of entries as $n$ increases, as we expect from Corollary 3. In Figure 1(b), we instead plot success probability against total number of observations per column. The fact that the curves coincide suggests that the samples per column, $m$, is constant with respect to $n$, which is precisely what Corollary 3 implies. Finally, in Figure 1(c), we rescale instead by $n/\log^2 n$, which corresponds to the passive sample complexity bound [21]. Empirically, the fact that these curves do not line up demonstrates that our algorithm requires fewer than $\log^2 n$ samples per column, outperforming the passive bound.

The second row of Figure 1 plots the same probability of success curves for the Singular Value Thresholding (SVT) algorithm [4]. As is apparent from the plots, SVT does not enjoy a linear dependence on $n$; indeed Figure 1(f) confirms the logarithmic dependency that we expect for passive matrix completion, and establishes that our algorithm has empirically better performance.

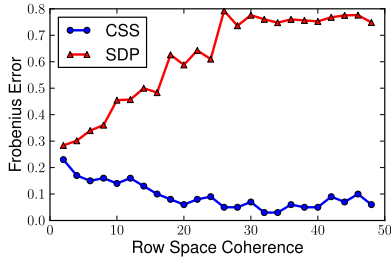

| | Unknown $M$ | | | Results |
| --- | --- | --- | --- | --- |
| $n$ | $r$ | $m/d_r$ | $m/n^2$ | time (s) |
| | 10 | 3.4 | 0.07 | 16 |
| 1000 | 50 | 3.3 | 0.33 | 29 |
| | 100 | 3.2 | 0.61 | 45 |
| | 10 | 3.4 | 0.01 | 3 |
| 5000 | 50 | 3.5 | 0.07 | 27 |
| | 100 | 3.4 | 0.14 | 104 |
| | 10 | 3.4 | 0.01 | 10 |
| 10000 | 50 | 3.5 | 0.03 | 84 |
| | 100 | 3.5 | 0.07 | 283 |

Figure 2: Reconstruction error as a function of row space incoherence for our noisy algorithm (CSS) and the semidefinite program of [20].

Table 1: Computational results on large low-rank matrices. $d_r = r(2n - r)$ is the degrees of freedom, so $m/d_r$ is the oversampling ratio.

In the third row, we study the algorithm's dependence on $r$ on $500 \times 500$ square matrices. In Figure 1(g) we plot the probability of success of the algorithm as a function of the sampling probability $p$ for matrices of various rank, and observe that the sample complexity increases with $r$. In Figure 1(h) we rescale the $x$-axis by $r^{-3/2}$ so that if our theorem is tight, the curves should coincide. In Figure 1(i) we instead rescale the $x$-axis by $r^{-1}$ corresponding to our conjecture about the performance of the algorithm. Indeed, the curves line up in Figure 1(i), demonstrating that empirically, the number of samples needed per column is linear in $r$ rather than the $r^{3/2}$ dependence in our theorem.

To confirm the computational improvement over existing methods, we ran our matrix completion algorithm on large-scale matrices, recording the running time and error in Table 1. To contrast with SVT, we refer the reader to Table 5.1 in [4]. As an example, recovering a $10000 \times 10000$ matrix of rank 100 takes close to 2 hours with the SVT, while it takes less than 5 minutes with our algorithm.

For the noisy algorithm, we study the dependence on row-space incoherence. In Figure 2, we plot the reconstruction error as a function of the row space coherence for our procedure and the semidefinite program of Negahban and Wainwright [20], where we ensure that both algorithms use the same number of observations. It's readily apparent that the SDP decays in performance as the row space becomes more coherent while the performance of our procedure is unaffected.

## 7 Conclusions and Open Problems

In this work, we demonstrate how sequential active algorithms can offer significant improvements in time, and measurement overhead over passive algorithms for matrix and tensor completion. We hope our work motivates further study of sequential active algorithms for machine learning.

Several interesting theoretical questions arise from our work:

1. Can we tighten the dependence on rank for these problems? In particular, can we bring the dependence on $r$ down from $r^{3/2}$ to linear? Simulations suggest this is possible.
2. Can one generalize the nuclear norm minimization program for matrix completion to the tensor completion setting while providing theoretical guarantees on sample complexity?

We hope to pursue these directions in future work.

### Acknowledgements

This research is supported in part by AFOSR under grant FA9550-10-1-0382 and NSF under grant IIS-1116458. AK is supported in part by a NSF Graduate Research Fellowship. AK would like to thank Martin Azizyan, Sivaraman Balakrishnan and Jayant Krishnamurthy for fruitful discussions.

## Footnotes

[1]We write $[n]$ for $\{1, \ldots, n\}$

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
