[Supplementary Material]

# A  Proof of Corollary 3

Corollary 3 is considerably simpler to prove than Theorem 2, so we prove the former in its entirety before proceeding to the latter. To simplify the presentation, a number of technical lemmas regarding incoherence and concentration of measure are deferred to sections E and F, respectively.

The proof begins by ensuring that for every column $c_j$, if $c_j \notin \tilde{U}$ then $||(I - \mathcal{P}_{\tilde{U}_\Omega})c_{j\Omega}||^2 > 0$ with high probability. This property is established in the following lemma:

**Lemma 7.** *Suppose that $\tilde{U} \subset U$ and a column $c_j \in U$ but $c_j \notin \tilde{U}$. If $m \geq 36r^{3/2}\mu_0 \log(2r/\delta)$ then with probability $\geq 1 - 4\delta$, $||(I - \mathcal{P}_{\tilde{U}_\Omega})c_{j\Omega}||^2 > 0$. If $c_j \in \tilde{U}$ then with probability $1$, $||(I - \mathcal{P}_{\tilde{U}_\Omega})c_{j\Omega}||^2 = 0$.*

*Proof of Lemma 7.* Decompose $c_j = x + v$ where $x \in \tilde{U}$ and $v \in \tilde{U}^\perp$. We can immediately apply Theorem 4 and are left to verify that the left hand side of Equation 5 is strictly positive. Since $c_j \notin \tilde{U}$ we know that $||v||_2^2 > 0$. Then:

$$\alpha = \sqrt{\frac{2\mu(v)}{m} \log(1/\delta)} + \frac{2\mu(v)}{3m} \log(1/\delta) \leq \sqrt{\frac{2r\mu_0}{m} \log(1/\delta)} + \frac{2r\mu_0}{3m} \log(1/\delta) < 1/2$$

When $m \geq 32r\mu_0 \log(1/\delta)$. Here we used that $\mu(v) \leq r\mu(U)$ since $v \in \text{span}(U)$. For $\gamma$:

$$\gamma = \sqrt{\frac{8d\mu(\tilde{U})}{3m} \log\left(\frac{2d}{\delta}\right)} \leq \sqrt{\frac{8r\mu_0}{3m} \log\left(\frac{2r}{\delta}\right)} \leq \frac{1}{3}$$

Whenever $m \geq 24r\mu_0 \log(2r/\delta)$. Finally, with the bounds on $\alpha$ and $\gamma$, the expression in Equation 5 is strictly positive when $3r\mu_0\beta \leq m$ since $d\mu(\tilde{U}) \leq r\mu_0$. Plugging in the definition of $\beta$ we require:

$$6\log(r/\delta) + \frac{4}{3}\frac{r^2\mu_0}{m} \log^2(r/\delta) \leq \frac{m}{3r\mu_0}$$

Which certainly holds when $m \geq 36r^{3/2}\mu_0 \log(r/\delta)$, concluding the proof. $\square$

It is easy to see that if $c_i \in \tilde{U}$ then $||(I - \mathcal{P}_{\tilde{U}_\Omega})c_{i\Omega}||^2 = 0$ deterministically and our algorithm does not further sample these columns. We must verify that these columns can be recovered exactly, and this amounts to checking that $\tilde{U}_\Omega^T \tilde{U}_\Omega$ is invertible. Fortunately, this was established as a lemma in [2], and in fact, the failure probability is subsumed by the probability in Theorem 4. Now we argue for correctness: there can be at most $r$ columns for which $||(I - \mathcal{P}_{\tilde{U}_\Omega})c_{\Omega i}||^2 > 0$ since $\text{rank}(M) \leq r$. For each of these columns, from Lemma 7, we know that with probability $1 - 4\delta$ $||(I - \mathcal{P}_{\tilde{U}_\Omega})c_{\Omega i}||^2 > 0$. By a union bound, with probability $\geq 1 - 4r\delta$ all of these tests succeed, so the subspace $\tilde{U}$ at the end of the algorithm is exactly the column space of $M$, namely $U$. All of these columns are recovered exactly, since we completely sample them.

The probability that the matrices $\tilde{U}_\Omega^T \tilde{U}_\Omega$ are invertible is subsumed by the success probability of Theorem 4, except for the last matrix. In other words, the success of the projection test depends on the invertibility of these matrices, so the fact that we recovered the column space $U$ implies that these matrices were invertible. The last matrix is invertible except with probability $\delta$ by Lemma 18.

If these matrices are invertible, then since $c_i \in \tilde{U}$, we can write $c_i = \tilde{U}\alpha_i$ and we have:

$$\hat{c}_i = \tilde{U}(\tilde{U}_\Omega^T \tilde{U}_\Omega)^{-1}\tilde{U}_\Omega^T \tilde{U}_\Omega \alpha_i = \tilde{U}\alpha_i = c_i$$

So these columns are all recovered exactly. This step only adds a factor of $\delta$ to the failure probability, leading to the final term in the failure probability of the theorem.

For the running time, per column, the dominating computational costs involve the projection $\mathcal{P}_{\tilde{U}_\Omega}$ and the reconstruction procedure. The projection involves several matrix multiplications and the inversion of a $r \times r$ matrix, which need not be recomputed on every iteration. Ignoring the matrix inversion, this procedure takes at most $O(n_1 r)$ per column for a total running time of $O(n_1 n_2 r)$. At most $r$ times, we must recompute $(U_\Omega^T U_\Omega)^{-1}$, which takes $O(r^2 m)$, contributing a factor of $O(r^3 m)$ to the total running time. Finally, we run the Gram-Schmidt process once over the course of the algorithm, which takes $O(n_1 r^2)$ time. This last factor is dominated by $n_1 n_2 r$.

# B  Proof of Theorem 2

We first focus on the recovery of the tensor in total, expressing this in terms of failure probabilities in the recursion. Then we inductively bound the failure probability of the entire algorithm. Finally, we compute the total number of observations. For now, define $\tau_T$ to be the failure probability of recovering a $T$-order tensor.

By Lemma 13, the subspace spanned by the mode-$T$ tensors has incoherence at most $r^{T-2}\mu_0^{T-1}$ and rank at most $r$ and each slice has incoherence at most $r^{T-1}\mu_0^{T-1}$. By the same argument as Lemma 7, we see that with $m \geq 36r^{T-1/2}\mu_0^{T-1}\log(2r/\delta)$ the projection test succeeds in identifying informative subtensors (those not in our current basis) with probability $\geq 1 - 4\delta$. With a union bound over these $r$ subtensors, the failure probability becomes $\leq 4r\delta + \delta$, not counting the probability that we fail in recovering these subtensors, which is $r\tau_{T-1}$.

For each order $T-1$ tensor that we have to recover, the subspace of interest has incoherence at most $r^{T-3}\mu^{T-2}$ and with probability $\geq 1 - 4r\delta$ we correctly identify each informative subtensor as long as $m \geq 36r^{T-3/2}\mu^{T-2}\log(2r/\delta)$. Again the failure probability is $\leq 4r\delta + \delta + r\tau_{T-2}$.

To compute the total failure probability we proceed inductively. $\tau_1 = 0$ since we completely observe any one-mode tensor (vector). The recurrence relation is:

$$\tau_t = 4r\delta + \delta + r\tau_{t-1} \tag{8}$$

which solves to:

$$\tau_T = \delta + 4r^{T-1}\delta + \sum_{t=1}^{T-2} 5r^t\delta \leq 5\delta T r^T \tag{9}$$

We also compute the sample complexity inductively. Let $m_T$ denote the number of samples needed to complete a $T$-order tensor. Then $m_1 = n_1$ and:

$$m_t = rm_{t-1} + 36n_t r^{t-1/2}\mu_0^{t-1}\log(2r/\delta) \tag{10}$$

So that $m_T$ is upper bounded as:

$$m_T = r^{T-1}n_1 + \sum_{t=2}^{T} r^{T-t}36n_t r^{t-1/2}\mu_0^{t-1}\log(2r/\delta) \leq 36(\sum_{t=1}^{T} n_t)r^{T-1/2}\mu_0^{T-1}\log(2r/\delta)$$

The running time is computed in a similar way to the matrix case. Assume that the running time to complete an order $t$ tensor is:

$$O(r(\prod_{i=1}^{t} n_i) + \sum_{i=2}^{t} m_i r^{3+t-i})$$

Note that this is exactly the running time of our Algorithm in the matrix case.

Per order $T - 1$ subtensor, the projection and reconstructions take $O(r\prod_{t=1}^{T-1} n_t)$, which in total contributes a factor of $O(r\prod_{t=1}^{T} n_t)$. At most $r$ times, we must complete an order $T-1$ subtensor, and invert the matrix $U_\Omega^T U_\Omega$. These two together take in total:

$$O\left(r\left[r(\prod_{t=1}^{T-1} n_t) + \sum_{t=2}^{T-1} m_t r^{3+T-1-t}\right] + r^3 m_T\right)$$

Finally the cost of the Gram-schmidt process is $r^2 \prod_{t=1}^{T-1} n_t$ which is dominated by the other costs. In total the running time is:

$$O\left(r\left(\prod_{t=1}^{T} n_t\right) + r^2 \prod_{t=1}^{T-1} n_t + \sum_{t=2}^{T} m_t r^{3+T-t}\right) = O\left(r\left(\prod_{t=1}^{T} n_t\right) + \sum_{t=2}^{T} m_t r^{3+T-t}\right)$$

since $r \leq n_T$. Now plugging in that $m_i = \tilde{O}(r^{2(i-1)})$, the terms in the second sum are each $\tilde{O}(r^{T+t+1})$ meaning that the sum is $\tilde{O}(r^{2T+1})$. This gives the computational result.

## C Proof of Theorem 6

We will first prove a more general result and obtain Theorem 6 as a simple consequence.

**Theorem 8.** *Let $M = A + R$ where $A = U\Sigma V^T$ and $R_{ij} \sim \mathcal{N}(0, \sigma^2)$. Let $M_r$ denote the best rank $r$ approximation to $M$. Assume that $A$ is rank $r$ and $\mu(U) \leq \mu_0$. For every $\delta, \epsilon \in (0, 1)$ sample a set of size $s = \frac{5Lr}{2\delta\epsilon}$ at each of the $L$ rounds of the algorithm and compute $\hat{M}$ as prescribed. Then with probability $\geq 1 - 9\delta$:*

$$||M - \hat{M}||_F^2 \leq 5/4 \left( \frac{1}{(1 - \epsilon)} ||M - M_r||_F^2 + \epsilon^L ||M||_F^2 \right)$$

*and the algorithm has expected sample complexity:*

$$\Omega \left( \frac{L^2 r}{\delta\epsilon} \left( n_1 + \mu_0 n_2 \sqrt{r} \log^2 \left( \frac{n_1 n_2 L r}{\delta\epsilon} \right) \right) \right)$$

The proof of this result involves some modifications to the analysis in [10]. We will follow their proof, allowing for some error in the sampling probabilities, and arrive at a recovery guarantee. Then we will show how these sampling probabilities can be well-approximated from limited observations.

The first Lemma analyzes a single round of the algorithm, while the second gives an induction argument to chain the first across all of the rounds. These are extensions of Theorems 2.1 and Theorems 1.2, respectively, from [10].

**Lemma 9.** *Let $M = U\Sigma V^T \in \mathbb{R}^{n_1 \times n_2}$ and let $\tilde{U}$ be a subspace of $\mathbb{R}^{n_1}$. Let $E = M - \mathcal{P}_{\tilde{U}} M$ and let $S$ be a random sample of $s$ columns of $M$, sampled according to the distribution $\hat{p}_i$ with:*

$$\frac{1 - \alpha_1}{1 + \alpha_2} \frac{||E_i||^2}{||E||_F^2} \leq \hat{p}_i \leq \frac{1 + \alpha_2}{1 - \alpha_1} \frac{||E_i||^2}{||E||_F^2}$$

*Then with probability $\geq 1 - \delta$ we have:*

$$||M - \mathcal{P}_{\tilde{U} \cup span(S), r} M||_F^2 \leq \frac{r}{s\delta} \frac{1 + \alpha_2}{1 - \alpha_1} ||E||_F^2 + ||M - M_r||_F^2$$

*Where $\mathcal{P}_{H,r}$ denotes a projection on to the best $r$-dimensional subspace of $H$ and $M_r$ is the best rank $r$ approximation to $M$.*

*Proof.* The proof closely mirrors that of Theorem 2.1 in [10]. The main difference is that we are using an estimate of the correct distribution, and this will result in some additional error.

For completeness we provide the proof here. We number the left (respectively right) singular vectors of $M$ as $u^{(j)}$ ($v^{(j)}$) and use subscripts to denote columns. We will construct $r$ vectors $w^{(1)}, \ldots, w^{(r)} \in \tilde{U} \cup \text{span}(S)$ and use them to upper bound the projection. In particular we have:

$$||M - \mathcal{P}_{\tilde{U} \cup \text{span}(S), r} M||_F^2 \leq ||M - \mathcal{P}_W M||_F^2$$

so we can exclusively work with $W$.

For each $i = 1, \ldots, n_2$ and for each $l = 1, \ldots s$ define:

$$X_l^{(j)} = \frac{1}{\hat{p}_i} E_i v_i^{(j)} \text{ with probability } \hat{p}_i$$

That is the $i$th column of the residual $E$, scaled by the $i$th entry of the $j$th right singular vector, and the sampling probability. Defining $X^{(j)} = \frac{1}{s} \sum_{l=1}^{s} X_l^{(j)}$, we see that:

$$\mathbb{E}[X^{(j)}] = \mathbb{E}[X_l^{(j)}] = \sum_{i=1}^{n_2} \frac{\hat{p}_i}{\hat{p}_i} E_i v_i^{(j)} = E v^{(j)}$$

Defining $w^{(j)} = \mathcal{P}_{\tilde{U}}(M) v^{(j)} + X^{(j)}$ and using the definition of $E$, it is easy to verify that $\mathbb{E}[w^{(j)}] = \sigma_j u^{(j)}$. It is also easy to see that $w^{(j)} - \sigma_j u^{(j)} = X^{(j)} - E v^{(j)}$.

We will now proceed to bound the second central moment of $w^{(j)}$.

$$\mathbb{E}[||w^{(j)} - \sigma_j u^{(j)}||^2] = \mathbb{E}[||X^{(j)}||^2] - ||Ev^{(j)}||^2$$

The first term can be expanded as:

$$\mathbb{E}[||X^{(j)}||^2] = \frac{1}{s^2} \sum_{l=1}^{s} \mathbb{E}[||X_l^{(j)}||^2] + \frac{s-1}{s}||Ev^{(j)}||^2$$

So that the second central moment is:

$$\mathbb{E}[||w^{(j)} - \sigma_j u^{(j)}||^2] = \frac{1}{s^2} \sum_{l=1}^{s} \mathbb{E}[||X_l^{(j)}||^2] - \frac{1}{s}||Ev^{(j)}||^2$$

Now we use the probabilities $\hat{p}_i$ to evaluate each term in the summation:

$$\mathbb{E}[||X_l^{(j)}||^2] = \sum_{i=1}^{n_2} \hat{p}_i \frac{||E^{(i)}v_i^{(j)}||^2}{\hat{p}_i^2} \leq \sum_{i=1}^{n_2} \frac{(1+\alpha_2)v_i^{(j)2}||E||_F^2}{1-\alpha_1} = \frac{1+\alpha_2}{1-\alpha_1}||E||_F^2$$

This gives us an upper bound on the second central moment:

$$\mathbb{E}[||w^{(j)} - \sigma_j u^{(j)}||^2] \leq \frac{1}{s}\frac{1+\alpha_2}{1-\alpha_1}||E||_F^2$$

To complete the proof, let $y^{(j)} = w^{(j)}/\sigma_j$ and define the matrix $F = (\sum_{j=1}^{k} y^{(j)}u^{(j)T})M$. Since $y^{(j)} \in W$, the column space of $F$ is contained in $W$ so $||M - \mathcal{P}_W(M)||_F^2 \leq ||M - F||_F^2$.

$$||M - F||_F^2 = \sum_{i=1}^{r} ||(M-F)v^{(i)}||^2 = \sum_{i=k+1}^{r} \sigma_i^2 + \sum_{i=1}^{k} ||\sigma_i u^{(i)} - w^{(i)}||^2$$

We now use Markov's inequality on the second term. Specifically, with probability $\geq 1-\delta$ we have:

$$||M - F||_F^2 \quad \leq \quad ||M - M_k||_F^2 + \frac{1}{\delta}\mathbb{E}[\sum_{i=1}^{k} ||\sigma_i u^{(i)} - w^{(i)}||^2] \leq ||M - M_r||_F^2 + \frac{r}{\delta s}\frac{1+\alpha_2}{1-\alpha_1}||E||_F^2$$

$\square$

**Lemma 10.** *Suppose that $(1+\alpha_2)/(1-\alpha_1) \leq c$ for some constant $c$ and for each of $L$ rounds of sampling. Let $S_1, \ldots, S_L$ denote the sets of columns selected at each round and set $s = \frac{Lcr}{\delta\epsilon}$. Then with probability $\geq 1-\delta$ we have:*

$$||M - \mathcal{P}_{\bigcup_{i=1}^{L} S_i, r}M||_F^2 \leq \frac{1}{1-\epsilon}||M - M_r||_F^2 + \epsilon^L||M||_F^2$$

*Proof.* The proof is by induction on the number of rounds $L$. We will have each round of the algorithm fail with probability $\delta/L$ so that the total failure probability will be at most $\delta$. The base case follows from Lemma 9. At the $l$th round, the same lemma tells us:

$$||M - \mathcal{P}_{\bigcup_{i=1}^{l} S_i, r}M||_F^2 \leq ||M - M_r||_F^2 + \frac{lcr}{s\delta}||E||_F^2$$

Plugging in our choice of $s$ and the definition of $E$:

$$||M - \mathcal{P}_{\bigcup_{i=1}^{l} S_i, r}M||_F^2 \leq ||M - M_r||_F^2 + \epsilon||M - \mathcal{P}_{\bigcup_{i=1}^{l-1} S_i, r}M||_F^2$$

and applying the induction hypothesis we have:

$$||M - \mathcal{P}_{\bigcup_{i=1}^{L} S_i, r}M||_F^2 \leq ||M - M_r||_F^2 + \epsilon(\frac{1}{1-\epsilon}||M - M_r||_F^2 + \epsilon^{L-1}||M||_F^2)$$

which gives us the desired result. $\square$

To complete the proof, we just need to compute how many observations are necessary to ensure that $(1 + \alpha_2)/(1 - \alpha_1) \leq c$. We can do this by manipulating Theorem 4 and upper bounding the incoherences of the subspaces throughout the execution of the algorithm.

**Lemma 11.** *We have:*

$$\frac{2}{5} \frac{||E_i||_2^2}{||E||_F^2} \leq \hat{p}_i \leq \frac{5}{2} \frac{||E_i||_2^2}{||E||_F^2}$$

*with probability $\geq 1 - 6\delta$ as long as the expected number of samples observed per column $m$ satisfies:*

$$m = \Omega\left(\frac{L^2 r^{3/2} \mu(U)}{\delta\epsilon} \log^2(n_1 n_2 L r/\delta\epsilon)\right)$$

*Proof.* To establish the result, we will use the concentration results from Section F and the incoherence results form Section E. The goal will be to apply Theorem 4 with a union bound across all rounds and all columns, but we first need to bound the incoherences.

With a union bound, Lemma 14 shows that each column (once projected onto the orthogonal complement of one of the subspaces) has incoherence $O(r\mu(U)\log(n_1 n_2 L/\delta))$ with probability $\geq 1-\delta$. At the same time, Lemma 15 reveals that with probability $\geq 1 - \delta$ all of the subspaces in the algorithm have incoherence at most $O(\mu(U)\log(n_1 L/\delta))$.

We can now apply Theorem 4. We will, as usual, take a union bound across all columns and all rounds, so each $\delta$ term in that lemma will be replaced with a $\delta/(n_1 L)$. Denote by $\tilde{U}_l$ the subspace projected onto during the $l$th round of the algorithm. With $m$ as in the lemma, the condition that $m \geq 8/3\dim(\tilde{U}_l)\mu(\tilde{U}_l)\log\left(\frac{2rn_1 L}{\delta}\right)$ is clearly satisfied, since $\dim(\tilde{U}_l) \leq \frac{L^2 r}{\delta\epsilon}$ and $\mu(\tilde{U}_l) \leq c\mu(U)\log(n_1 L/\delta)$. We also have that:

$$
\begin{aligned}
\alpha &= \sqrt{\frac{2\mu(v)}{m}\log(\frac{n_1 L}{\delta})} + \frac{2}{3}\frac{\mu(v)}{m}\log(n_1 L/\delta) \\
&\leq c_1\sqrt{\frac{r\mu(U)\log^2(n_1 n_2 L/\delta)}{m}} + c_2\frac{r\mu(U)\log^2(n_1 n_2 L/\delta)}{m} \leq O(1)
\end{aligned}
$$

By boosting the size of $m$ by a constant, we can make $\alpha \leq 1/4$. For $\gamma$ we have:

$$\gamma = \sqrt{\frac{8\dim(\tilde{U}_l)\mu(\tilde{U}_l)}{3m}\log(2\dim(\tilde{U}_l)/\delta)} \leq c\sqrt{\frac{L^2 k}{\delta\epsilon}\frac{\mu(U)}{m}\log^2(\frac{n_1 r L^3}{\delta^2\epsilon})} \leq 1/3$$

if we choose the constants correctly. Finally we have:

$$
\begin{aligned}
\beta &= 6\log(n_1 L\dim(\tilde{U}_l)/\delta) + \frac{4}{3}\frac{\dim(\tilde{U}_l)\mu(v)}{m}\log^2(n_1 L\dim(\tilde{U}_l)/\delta) \\
&\leq \log(\frac{n_1 r L^3}{\delta^2\epsilon}) + \frac{L^2 r^2 \mu(U)}{m\delta\epsilon}\log^3(\frac{n_1 n_2 r L^3}{\delta^2\epsilon})
\end{aligned}
$$

which gives:

$$\frac{\dim(\tilde{U}_l)\mu(\tilde{U}_l)}{m}\frac{\beta}{(1-\gamma)} \leq \frac{L^2 r\mu(U)}{m\delta\epsilon}\log^2(\frac{n_1 r L^3}{\delta^2\epsilon}) + \frac{L^4 r^3 \mu(U)^2}{m^2\delta^2\epsilon^2}\log^4(\frac{n_1 n_2 r L^3}{\delta^2\epsilon}) \leq O(1)$$

again using our definition of $m$. In particular, if we make this bound $\leq 1/4$ we then have that:

$$\frac{m}{n_1}(1 - 1/2)||v - \mathcal{P}_S v||_2^2 \geq ||v_\Omega - \mathcal{P}_{S_\Omega} v_\Omega||_2^2 \geq \frac{m}{n_1}(1 + 1/4)||v - \mathcal{P}_S v||_2^2$$

in which case:

$$\hat{p}_i = \frac{||v_{i\Omega} - \mathcal{P}_{S_\Omega} v_{i\Omega}||_2^2}{\sum_i ||v_{i\Omega} - \mathcal{P}_{S_\Omega} v_{i\Omega}||_2^2} \leq \frac{5}{2}p_i$$

along with the other direction. □

We are essentially done proving the theorem. The total number of samples used is:

$$n_2 m = \Omega \left( \frac{n_2 L^2 r^{3/2} \mu(U)}{\delta \epsilon} \log^3 \left( \frac{n_1 n_2 L r}{\delta \epsilon} \right) \right)$$

We also completely observe $\Omega(L^2 r / \delta \epsilon)$ columns. In total this gives us the sample complexity bound in Theorem 8. The failure probability is $\leq 7\delta$ ($6\delta$ from Lemma 11 and $\delta$ from Lemma 10).

So far we have recovered a subspace that can be used to approximate $M$. Unfortunately, we cannot actually compute $\mathcal{P}_{U_L} M$ given limited samples. Instead, for each column $c$, we compute $\hat{c} = U_L (U_{L\Omega}^T U_{L\Omega})^{-1} U_{\Omega L} c_\Omega$ and use $\hat{c}$ as our estimate of the column. This is similar to another projection operation, and the error will only be a constant factor worse than before.

**Lemma 12.** *Let $c_i$ denote a column of the matrix $M$ and let $\hat{U}$ denote the subspace at the end of the adaptive algorithm. Write $\hat{c} = \hat{U}(\hat{U}_\Omega^T \hat{U}_\Omega)^{-1} \hat{U}_\Omega c$ Then with probability $\geq 1 - 2\delta$:*

$$||c - \hat{c}||^2 \leq \left( 1 + \frac{r \mu(\hat{U}) \beta}{m(1-\gamma)^2} \right) ||\mathcal{P}_{\hat{U}^\perp} c||^2$$

*With $\beta$ and $\gamma$ defined as in Theorem 4.*

*Proof.* Decompose $c = x + y$ where $x \in \hat{U}$ and $y \in \hat{U}^\perp$. It's easy to see that $x = \hat{U}(\hat{U}_\Omega^T \hat{U}_\Omega)^{-1} \hat{U}_\Omega x_\Omega$ so we are left with:

$$||y - \hat{U}(\hat{U}_\Omega^T \hat{U}_\Omega)^{-1} \hat{U}_\Omega y||^2 = ||y||^2 + ||\hat{U}(\hat{U}_\Omega^T \hat{U}_\Omega)^{-1} \hat{U}_\Omega y||^2$$

Because $y \in U^\perp$ so the cross term is zero. The second term here is equivalant to:

$$||(\hat{U}_\Omega^T \hat{U}_\Omega)^{-1} \hat{U}_\Omega y||^2 \leq ||(\hat{U}_\Omega^T \hat{U}_\Omega)^{-1}||_2^2 ||\hat{U}_\Omega y||_2^2$$

By Lemma 3 in [2] the first term is upper bounded by $\frac{n_1^2}{(1-\gamma)^2 m^2}$ while Lemma 17 reveals that the second term is upper bounded by $\beta \frac{m}{n_1^2} r \mu(\hat{U}) ||y||^2$. Combining these two yields the result. $\square$

We already showed that with our choice of $m$, the expression in the above Lemma is smaller than $5/4$. Moreover the probability of failure simply adds $2\delta$ to the total failure probability. Thus:

$$||M - \hat{M}||_F^2 = \sum_i ||c_i - \hat{c}_i||_2^2 \leq 5/4 ||M - \mathcal{P}_{U_T} M||_F^2$$

and the last expression we bounded previously.

## C.1 Proving the Theorem

To prove the main theorem, it is best to view $M$ as equal to $A$ on all of the unobserved entries. In other words, if $\Omega$ is the set of all observations over the course of the algorithm, the random matrix $R$ is zero on $\Omega^C$. Since we never observed $M$ on $\Omega^C$, we have no way of knowing whether $M$ was equal to $A$ on those coordinates. It is therefore fair to write $M = A + R_\Omega$ where $R$ is zero on $\Omega^C$.

We expand the norm and then apply the main theorem:

$$||A - \hat{M}||_F^2 \leq 3||M - \hat{M}||_F^2 + 3||R_\Omega||_F^2 \leq \frac{5}{4} \left( \frac{3}{1-\epsilon} ||M - M_r||_F^2 + 3\epsilon^L ||M||_F^2 \right) + 3||R_\Omega||_F^2$$

Now since $M_r$ is the best rank $r$ approximation to $M$ (in Frobenius norm) and since $A$ is rank $r$, we know that $||M - M_r||_F \leq ||M - A||_F$. With this substitution and setting $\epsilon = 1/2$, $L = \log_2(n_1 n_2)$ we will arrive at the result (below constants are denoted by $c$ and they change from line to line):

$$||A - \hat{M}||_F^2 \quad \leq \quad c_1 ||M - A||_F^2 + \frac{c_2}{n_1 n_2} ||M||_F^2 + c_3 ||R_\Omega||_F^2 \leq \frac{c_1}{n_1 n_2} ||A||_F^2 + c_2 ||R_\Omega||_F^2$$

which holds as long as $n_1 n_2$ is sufficiently large.

# D  Proof of Theorem 5

We start by giving a proof in the matrix case, which is a slight variation of the proof by Candes and Tao [7]. Then we turn to the tensor case, where only small adjustments are needed to establish the result. We work in the Bernoulli model, noting that Candes' and Tao's arguments demonstrate how to adapt these results to the uniform-at-random sampling model.

## D.1  Matrix Case

In the matrix case, suppose that $l_1 = \frac{n_1}{r}$ and $l_2 = \frac{n_2}{\mu_0 r}$ are both integers. Define the following blocks $R_1, \dots R_r \subset [n_1]$ and $C_1, \dots C_r \subset [n_2]$ as:

$$
\begin{aligned}
R_i &= \{l_1(i-1)+1, l_1(i-1)+2, \dots l_1 i\} \\
C_i &= \{l_2(i-1)+1, l_2(i-1)+2, \dots l_2 i\}
\end{aligned}
$$

Now consider the $n_1 \times n_2$ family of matrices defined by:

$$
\mathcal{M} = \{\sum_{k=1}^{r} u_k v_k^T | u_k = [1, \sqrt{\mu_0}]^n \otimes \mathbf{1}_{R_k}, v_k = \mathbf{1}_{C_k}\} \tag{11}
$$

$\mathcal{M}$ is a family of block-diagonal matrices where the blocks have size $l_1 \times l_2$. Each block has constant rows whose entries may take arbitrary values in $[1, \sqrt{\mu_0}]$. For any $M \in \mathcal{M}$, the incoherence of the column space can be computed as:

$$
\mu(U) = \frac{n_1}{r} \max_{j \in [n_1]} ||\mathcal{P}_U e_j||_2^2 = \frac{n_1}{r} \max_{k \in [r]} \max_{j \in [n_1]} \frac{(u_k^T e_j)^2}{(u_k^T u_k)^2} \leq \frac{n_1}{r} \max_{k \in [r]} \frac{\mu_0}{(n_1/r)} = \mu_0
$$

A similar calculation reveals that the row space is also incoherent with parameter $\mu_0$.

Unique identification of $M$ is not possible unless we observe at least one entry from each row of each diagonal block. If we did not, then we could vary that corresponding coordinate in the appropriate $u_k$ and find infinitely many matrices $M' \in \mathcal{M}$ that agree with our observations, have rank and incoherence at most $r$ and $\mu_0$ respectively. Thus, the probability of successful recovery is no larger than the probability of observing one entry of each row of each diagonal block.

The probability that any row of any block is unsampled is $\pi_1 = (1-p)^{l_2}$ and the probability that all rows are sampled is $(1 - \pi_1)^{n_1}$. This must upper bound the success probability $1 - \delta$. Thus:

$$
-n_1 \pi_1 \geq n_1 \log(1 - \pi_1) \geq \log(1 - \delta) \geq -2\delta
$$

or $\pi_1 \leq 2\delta/n_1$ as long as $\delta < 1/2$. Substituting $\pi_1 = (1-p)^{l_2}$ we obtain:

$$
\log(1-p) \leq \frac{1}{l_2} \log\left(\frac{2\delta}{n_1}\right) = \frac{\mu_0 r}{n_2} \log\left(\frac{2\delta}{n_1}\right)
$$

as a necessary condition for unique identification of $M$.

Exponentiating both sides, writing $p = \frac{m}{n_1 n_2}$ and the fact that $1 - e^{-x} > x - x^2/2$ gives us:

$$
m \geq n_1 \mu_0 r \log\left(\frac{n_1}{2\delta}\right)(1 - \epsilon/2)
$$

when $\mu_0 r/n_2 \log(\frac{n_1}{2\delta}) \leq \epsilon < 1$.

## D.2  Tensor Case

Fix $T$, the order of the tensor and suppose that $l_1 = \frac{n_1}{r}$ is an integer. Moreover, suppose that $l_t = \frac{n_t}{\mu_0 r}$ is an integer for $1 < t \leq T$. Define a set of blocks, one for each mode and the family

$$
\begin{aligned}
B_i^{(t)} &= \{l_t(i-1)+1, l_t(i-1)+2, \dots, l_t i\} \forall i \in [r], t \in [p] \\
\mathcal{M} &= \left\{\sum_{i=1}^{r} \otimes_{t=1}^{T} a_i^{(t)} \middle| \begin{array}{l} a_i^{(1)} = [1, \sqrt{\mu_0}]^n \otimes \mathbf{1}_{B_i^{(t)}} \\ a_i^{(t)} = \mathbf{1}_{B_i^{(t)}}, 1 < t \leq T \end{array} \right\}
\end{aligned}
$$

This is a family of block-diagonal tensors and just as before, straightforward calculations reveal that each subspace is incoherent with parameter $\mu_0$. Again, unique identification is not possible unless we observe at least one entry from each row of each diagonal block. The difference is that in the tensor case, there are $\prod_{i\neq 1} l_i$ entries per row of each diagonal block so the probability that any single row is unsampled is $\pi_1 = (1-p)^{\prod_{i\neq 1} l_i}$. Again there are $n_1$ rows and any algorithm that succeeds with probability $1 - \delta$ must satisfy:

$$-n_1\pi_1 \geq n_1 \log(1 - \pi_1) \geq \log(1 - \delta) \geq -2\delta$$

Which implies $\pi_1 \leq 2\delta/n_1$ (assuming $\delta < 1/2$). Substituting in the definition of $\pi_1$ we have:

$$\log(1 - p) \leq \frac{1}{\prod_{i\neq j} l_i} \log\left(\frac{2\delta}{n_1}\right) = \frac{\mu_0^{T-1} r^{T-1}}{\prod_{i\neq j} n_i} \log\left(\frac{2\delta}{n_1}\right)$$

The same approximations as before yield the bound (as long as $\frac{\mu_0^{T-1} r^{T-1}}{\prod_{i\neq j} n_i} \log(\frac{n_1}{2\delta}) \leq \epsilon < 1$):

$$m \geq n_1 \mu_0^{T-1} r^{T-1} \log\left(\frac{n_1}{2\delta}\right)(1 - \epsilon/2)$$

## E  Properties about Incoherence

A significant portion of our proofs revolve around controlling incoherences of various subspaces used throughout the execution of the algorithms The following technical lemmas will enable us to work with these quantities.

**Lemma 13.** *Let* $U_1 \subset \mathbb{R}^{n_1}, U_2 \subset \mathbb{R}^{n_2}, \ldots U_T \subset \mathbb{R}^{n_T}$ *be subspaces of dimension at most d, let* $W_1 \subset U_1$ *have dimension* $d'$. *Define* $\mathbb{S} = span(\{\otimes_{t=1}^T u_i^{(t)}\}_{i=1}^d)$. *Then:*

*(a)* $\mu(W_1) \leq \frac{dim(U_1)}{d'} \mu(U_1)$.

*(b)* $\mu(\mathbb{S}) \leq d^{T-1} \prod_{i=1}^T \mu(U_i)$.

*Proof.* For the first property, since $W_1$ is a subspace of $U_1$, $\mathcal{P}_{W_1} e_j = \mathcal{P}_{W_1} \mathcal{P}_{U_1} e_j$ so $||\mathcal{P}_{W_1} e_j||_2^2 \leq ||\mathcal{P}_{U_1} e_j||_2^2$. The result now follows from the definition of incoherence.

For the second property, we instead compute the incoherence of:

$$\mathbb{S}' = span\left(\left\{\otimes_{t=1}^T u^{(t)}\right\}_{u^{(t)} \in U_t \forall t}\right)$$

which clearly contains $\mathbb{S}$. Note that if $\{u_i^{(t)}\}$ is an orthonormal basis for $U_t$ (for each $t$), then the outer product of all combinations of these vectors is a basis for $\mathbb{S}'$. We now compute:

$$
\begin{aligned}
\mu(\mathbb{S}') &= \frac{\prod_{i=1}^T n_i}{\prod_{t=1}^T \dim(U_t)} \max_{k_1 \in [n_1],\ldots,k_T \in [n_T]} ||\mathcal{P}_{\mathbb{S}'}(\otimes_{t=1}^T e_{k_t})||^2 \\
&= \frac{\prod_{i=1}^T n_i}{\prod_{t=1}^T \dim(U_t)} \max_{k_1,\ldots,k_T} \sum_{i_1,\ldots,i_T} \langle \otimes_{t=1}^T u_{i_t}^{(t)}, \otimes_{t=1}^T e_{k_t}\rangle^2 \\
&= \frac{\prod_{i=1}^T n_i}{\prod_{t=1}^T \dim(U_t)} \max_{k_1,\ldots,k_T} \sum_{i_1,\ldots,i_T} \prod_{t=1}^T (u_{i_t}^{(t)T} e_{k_t})^2 \\
&= \frac{\prod_{i=1}^T n_i}{\prod_{t=1}^T \dim(U_t)} \prod_{j=1}^T \max_{k_j} \sum_{i=1}^r (u_i^{(t)T} e_{k_j})^2 \leq \prod_{t=1}^T \mu(U_t)
\end{aligned}
$$

Now, property (a) establishes that $\mu(\mathbb{S}) \leq \frac{r^T}{r} \mu(\mathbb{S}')$ which is the desired result. $\qquad\square$

**Lemma 14.** *Let $U$ be the column space of $M$ and let $V$ be some other subspace of dimension at most $n_1/2 - k$. Let $v_i = \mathcal{P}_{V^\perp} c_i$ for each column $c_i$. Then with probability $\geq 1 - \delta$:*

$$\max_i \mu(v_i) \leq 3k\mu(U) + 24\log(2n_1 n_2/\delta) = O(k\mu(U)\log(n_1 n_2/\delta))$$

*Proof.* Decompose $v_i = x_i + r_i$ where $x_i \in U \cap V^\perp$ and $r_i \in U^\perp \cap V^\perp$. Since each column is composed of a deterministic component living in $U$ and a random component, it must be the case that $r_i$ is a random gaussian vector living in $U^\perp \cap V^\perp$, which is a subspace of dimension at least $n_1 - d \geq n_1 - \dim(U) - \dim(V)$. We can now proceed with the bound:

$$
\begin{aligned}
\mu(v_i) &= n_1 \frac{||v_i||_\infty^2}{||v_i||_2^2} \leq 3n_1 \frac{||x_i||_\infty^2 + ||r_i||_\infty^2}{||x_i||_2^2 + ||r_i||_2^2} \\
&\leq 3n_1 \frac{||x_i||_\infty^2}{||x_i||_2^2} + 3n_1 \frac{||r_i||_\infty^2}{||r_i||_2^2} \\
&\leq 3k\mu(U) + \frac{6\sigma^2 n_1 \log(2n_1 n_2/\delta)}{\sigma^2(n_1 - d) - 2\sigma^2\sqrt{(n_1 - d)\log(n_2/\delta)}}
\end{aligned}
$$

For the second line, we used that $\frac{\sum_i a_i}{\sum_i b_i} \leq \sum_i \frac{a_i}{b_i}$ whenever $a_i, b_i \geq 0$ which is the case here. Finally we use Lemma 19 on the denominator, 20 on the numerator, and a union bound over all $n_2$ columns. For $(n_1 - d)$ sufficiently large (as long as $\sqrt{(n_1 - d)\log n_2/\delta} \leq (n_1 - d)/4$) and if $d \leq n_1/2$ we can bound as:

$$3k\mu(U) + \frac{12n_1 \log(2n_1 n_2/\delta)}{n_1 - d} \leq 3k\mu(U) + 24\log(2n_1 n_2/\delta)$$

$\square$

**Lemma 15.** *Let $I_l = \bigcup_{i=1}^l S_i$ and let $U_l = span(\{c_i\}_{i \in I_l})$ as in the execution of the noisy algorithm. If $|I_l| \leq n_1/2$ then with probability $\geq 1 - \delta$, for all $l \in [L]$, we have:*

$$\mu(U_l) = O(\mu(U)\log(n_1 L/\delta))$$

*Proof.* It is clear that $U_l \subset span(\{c_i\}_{i \in I_l}) \bigcup span(\{r_i\}_{i \in I_l})$ which will make things much easier to analyze. Note that $span(\{c_i\}_{i \in I_l}) \subset U$ the original incoherent subspace and let $R_{I_l}$ denote the random matrix of columns corresponding to $I_l$. We then have:

$$
\begin{aligned}
||\mathcal{P}_{U_l} e_i||_2^2 &\leq ||\mathcal{P}_U e_i||_2^2 + ||\mathcal{P}_{U^\perp} \mathcal{P}_{R_{I_l}} e_i||_2^2 \leq ||\mathcal{P}_U e_i||_2^2 + ||\mathcal{P}_{R_{I_l}} e_i||_2^2 \\
&\leq \frac{r\mu(U)}{n_1} + ||R_{I_l}||_2^2 ||(R_{I_l}^T R_{I_l})^{-1}||_2^2 ||R_{I_l}^T e_i||_2^2 \\
&\leq \frac{r\mu(U)}{n_1} + \frac{(\sqrt{n_1} + \sqrt{|I_l|} + \epsilon)^2}{(\sqrt{n_1} - \sqrt{|I_l|} - \epsilon)^4}(|I_l| + 2\sqrt{|I_l|\log(1/\delta)} + 2\log(1/\delta))
\end{aligned}
$$

Now if $|I_l| \leq n_1/2$ and $\delta$ is not exponentially small, the contribution from the random matrix is:

$$\frac{(\sqrt{n_1} + \sqrt{|I_l|} + \sqrt{2\log(2/\delta)})^2}{(\sqrt{n_1} - \sqrt{|I_l|} - \sqrt{2\log(2/\delta)})^4}(|I_l| + 2\sqrt{|I_l|\log(1/\delta)} + 2\log(1/\delta)) = O(|I_l|\log(1/\delta)/n_1)$$

So the total incoherence will be (note that $\dim(U_l) = |I_l|$ with probability 1 since $|I_l| \leq n_1/2$):

$$\mu(U_l) = \frac{n_1}{|I_l|} \max_i ||\mathcal{P}_{U_l} e_i||_2^2 \leq \frac{n}{|I_l|}\left(\frac{r\mu(U)}{n} + O(|I_l|/n)\right) = O(\frac{r}{|I_l|}\mu(U) + 1) = O(\mu(U)\log(1/\delta))$$

The failure probability here is $\delta n_1 L$ if there are $L$ rounds so the incoherence is:

$$\mu(U_l) = O(\mu(U)\log(n_1 L))$$

$\square$

# F   A Collection of Concentration Results

We enumerate several concentration of measure lemmas that we use throughout our proofs. Many of these are well known results and we provide the references to their proofs.

## F.1   Proof of Theorem 4

We improve on the result of Balzano *et al.* [2] to establish Theorem 4. The proof parallels theirs but with improvements to two key Lemmas. The improvement stems from using Bernstein's inequality in lieu of standard Chernoff bounds in the concentration arguments and carries over into our sample complexity guarantees. Here we state and prove the two lemmas and then sketch the overal proof.

**Lemma 16.** *With the same notations as Theorem 4, with probability $\geq 1 - 2\delta$.*

$$(1-\alpha)\frac{m}{n}||v||_2^2 \leq ||v_\Omega||_2^2 \leq (1+\alpha)\frac{m}{n}||v||_2^2 \tag{12}$$

*Proof.* The difference between Lemma 16 and Lemma 1 from [2] is in the definition of $\alpha$. Here we have reduced the relationship between $\mu(v)$ and $m$ from $\mu(v)^2/m$ to $\mu(v)/m$. The proof is an application of Bernstein's inequality.

Let $X_i = v_{\Omega(i)}^2$ so that $\sum_{i=1}^m X_i = ||v_\Omega||_2^2$. We can compute the variance and bound for $X_i$ as:

$$\sigma^2 = \mathbb{E}[X_i^2] = \frac{1}{n}\sum_{i=1}^n v_i^4 \leq \frac{1}{n}||v||_\infty^2||v||_2^2, \quad M = \max|X_i| \leq ||v||_\infty^2$$

Now we apply Berstein's inequality:

$$\mathbb{P}\left(\left|\sum_{i=1}^m X_i - \mathbb{E}[\sum_{i=1}^m X_i]\right| > t\right) \leq 2\exp\left(\frac{1}{2}\frac{-t^2}{m\sigma^2 + \frac{1}{3}Mt}\right)$$

Noting that $\mathbb{E}[\sum_{i=1}^m X_i] = \frac{m}{n}||v||_2^2$ and setting $t = \alpha\frac{m}{n}||v||_2^2$ the bound becomes:

$$\mathbb{P}\left(\left|\sum_{i=1}^m X_i - \frac{m}{n}||v||_2^2\right| > \alpha\frac{m}{n}||v||_2^2\right) \leq 2\exp\left(\frac{-\alpha^2 m||v||_2^2}{2n||v||_\infty^2(1+\alpha/3)}\right) \leq 2\exp\left(\frac{-\alpha^2 m}{2\mu(v)(1+\alpha/3)}\right)$$

Finally plugging in the definition of $\alpha$ from the theorem shows that the right hand side is $\leq 2\delta$.  $\square$

In similar spirit to Lemma 16 we can also improve Lemma 2 from [2] using Bernstein's inequality:

**Lemma 17.** *With the same notations as Theorem 4, with probability at least $1 - \delta$:*

$$||U_\Omega^T v_\Omega||_2^2 \leq \beta\frac{m}{n}\frac{d\mu(U)}{n}||v||_2^2 \tag{13}$$

*Proof.* Again the improvement in our Lemma is in the expression $\beta$ where we have an improved dependence between $m$ and $\mu(y)$. The proof is an application of Bernstein's inequality. Note that:

$$||U_\Omega^T v_\Omega||_2^2 = \sum_{j=1}^d\left(\sum_{i\in\Omega}u_{ji}v_i\right)^2 = \sum_{j=1}^d\left(\sum_{i\in\Omega}X_{ji}\right)^2$$

Where we have defined $X_{ji} = \sum_{k=1}^n u_{jk}v_j\mathbf{1}_{\Omega(i)=k}$. We have:

$$\mathbf{E}[X_{ji}] = 0, \ \mathbf{E}[X_{ji}^2] = \frac{1}{n}\sum_{k=1}^n(u_{jk}v_k)^2 \triangleq \sigma_j^2, \ |X_{ji}| \leq ||u_j||_\infty||v||_\infty \triangleq M$$

We apply Bernstein's inequality and take a union bound, so that with probability $\geq 1 - \delta$:

$$\forall j = 1, \ldots, d \quad \sum_{i=1}^{m} X_{ji} \quad \leq \quad \sqrt{2m\sigma_j^2 \log(d/\delta)} + \frac{2}{3} M \log(d/\delta)$$

$$\sum_{j=1}^{d} \left( \sum_{i=1}^{m} X_{ji} \right)^2 \quad \leq \quad 3 \left( 2m \left( \sum_{j=1}^{d} \sigma_j^2 \right) \log(d/\delta) + \frac{4}{9} dM^2 \log^2(d/\delta) \right)$$

Notice that:

$$\sum_{j=1}^{d} \sigma_j^2 = \frac{1}{n} \sum_{j=1}^{d} \sum_{i=1}^{n} u_{ji} v_i \leq \frac{1}{n} \sum_{i=1}^{n} v_i^2 \sum_{j=1}^{d} u_{ji}^2 \leq \frac{1}{n} ||v||_2^2 d\mu(U)/n$$

Notice also that $||u_j||_\infty^2 \leq d\mu(U)/n$. Plugging in these bounds, with probability $\geq 1 - \delta$:

$$||U_\Omega^T v_\Omega||_2^2 \quad \leq \quad 3 \left( 2 \frac{m}{n} \frac{d\mu(U)}{n} ||v||_2^2 \log(d/\delta) + \frac{4}{9} \frac{d\mu(U)}{n} d||v||_\infty^2 \log^2(d/\delta) \right)$$

$$\leq \quad \frac{m}{n} \frac{d\mu(U)}{n} ||v||_2^2 \left( 6 \log(d/\delta) + \frac{4}{3} \frac{d\mu(v)}{m} \log^2(d/\delta) \right)$$

Where we used that $||v||_\infty^2 \leq ||v||_2^2 \mu(v)/n$ via the definition of incoherence. $\qquad \square$

It will also be essential for these projections matrices to be invertible even with missing observations, as this will allow us to reconstruct columns of the matrix.

**Lemma 18** ( [2]). *Let $\delta > 0$ and $m \geq \frac{8}{3} r\mu_0 \log(2r/\delta)$, Then:*

$$||(U_\Omega^T U_\Omega)^{-1}||_2 \leq \frac{n}{(1-\gamma)m} \tag{14}$$

*with probability $\geq 1 - \delta$, provided that $\gamma < 1$. In particular $U_\Omega^T U_\Omega$ is invertible.*

*Proof of Theorem 4.* Let $W_\Omega^T W_\Omega = (U_\Omega^T U_\Omega)^{-1}$. If Lemma 18 holds, $U_\Omega^T U_\Omega$ is invertible so

$$v_\Omega^T U_\Omega (U_\Omega^T U_\Omega)^{-1} U_\Omega^T v_\Omega = ||W_\Omega U_\Omega^T v_\Omega||_2^2 \leq ||W_\Omega||_2^2 ||U_\Omega^T v_\Omega||_2^2 \leq ||(U_\Omega^T U_\Omega)^{-1}|| ||U_\Omega^T v_\Omega||_2^2$$

And therefore:

$$||v_\Omega - \mathcal{P}_{U_\Omega} v_\Omega||_2^2 = ||v_\Omega||_2^2 - v_\Omega^T U_\Omega (U_\Omega^T U_\Omega)^{-1} U_\Omega^T v_\Omega \geq (1 - \alpha) \frac{m}{n} ||v||_2^2 - \frac{d\mu(U)}{n} \frac{\beta}{1-\gamma} ||v||_2^2$$

yields the lower bound. The upper bound follows from the same decomposition and Lemma 16. $\qquad \square$

## F.2 Concentration for Gaussian Vectors and Matrices

We will also need several concentration results pertaining to gaussian random vectors and gaussian random matrices. The first of these will help us bound the $\ell_2$ norm of a gaussian vector:

**Lemma 19.** *[19] Let $X \sim \chi_d^2$. Then with probability $\geq 1 - 2\delta$:*

$$-2\sqrt{d\log(1/\delta)} \leq X - d \leq 2\sqrt{d\log(1/\delta)} + \log(1/\delta)$$

A gaussian random vector $r$ has incoherence that depends on $||r||_\infty^2$ so it is crucial that we can control the maximum of gaussian random variables.

**Lemma 20.** *Let $X_1, \ldots, X_n \sim \mathcal{N}(0, \sigma^2)$. Then with probability $\geq 1 - \delta$:*

$$\max_i |X_i| \leq \sigma \sqrt{2 \log(2n/\delta)}$$

Finally, we will be projecting on to perturbed subspaces so we will need to control the coherence of these subspaces. The spectrum of the perturbation, will play a role in the coherence calculations.

**Lemma 21.** *[24] Let $R$ be a $n \times t$ whose entries are independent standard normal random variables. Then for every $\epsilon \geq 0$, with probability $1 - 2\exp\{-\epsilon^2/2\}$, one has:*

$$\sqrt{n} - \sqrt{t} - \epsilon \leq \sigma_{\min}(R) \leq \sigma_{\max}(R) \leq \sqrt{n} + \sqrt{t} + \epsilon$$