[Reviews · NeurIPS 2013]

Submitted by Assigned_Reviewer_4

The authors propose sequential and active formulations for low rank matrix completion including a formulation where the underlying matrix/tensor is known to be noisy. The resultant formulations have strong performance bounds with regard to the number of entries that need to be polled as a function of the matrix rank.

Theorem 3 is in my opinion a main contribution and it was good to see it was experimentally verified. The lower bound for uniform sampling given in theorem 4 is also useful, but it appears you did not experimentally verify how close your work is to this lower bound. Please correct me if I'm wrong.

The experimental results are solid and I presume limited to simulated results due to availability of real data. The comparison to SVT shows a clear time improvement though I don't think you could draw the conclusion it is always an order of magnitude faster given your results.
Summary: A very interesting problem with a novel problem setting, solid theoretical bounds that are empirically verified. The only down side to the paper is the lack of experimental results on real world problems, but that is a minor issue in my opinion.

Submitted by Assigned_Reviewer_6

This paper provides algorithms and results for low rank tensor completion under adaptive sampling. The authors propose algorithms with following guarantees:
1. For the case of completing a $T^{th}$ order tensor of rank $r$, $O(r^{2T-2} T^2 n \log r)$ samples suffice.
2. Specializing the above for matrix completion yields $O(r^2 n \log r)$ sample complexity.
3. For noisy matrix completion (under Gaussian noise), $O(r^2 n polylog(n))$ samples suffice.

The authors further show that their approach (theoretically) outperforms any passive sampling scheme.

Quality: The main contribution of the paper is an algorithm that completes a rank-r T^th order tensor of dimension $nxnx...n$ using adaptive sampling and has a provable sample complexity of $r^O(T) O(n)$. The key take away from here is that the authors obtain optimal dependence in the tensor dimensions, though the dependence is exponential in the tensor order. The key insight in the paper is the fact that inner products between two (incoherent) vectors can be estimated by sampling. However, this idea is not new and has been established in prior work by Balzano et. al. (as mentioned in the paper by the authors). The authors then establish incoherence of tensor modes and the analysis uses standard techniques.

For the noisy case, the authors adapt the algorithm of Deshpande et. al. which samples vectors using inner products. Since the inner products are not known in the current case, the authors propose sampling to estimate inner products, and then apply Deshpande et. al.'s algorithm.

Clarity: The paper is well written and easy to follow.

Originality: The ideas used in the paper are by now standard in the literature. However, the authors use them in a new setting.

Significance: After reading the author feedback, I agree that obtaining the right dependence on the matrix dimensions is an important step. However, in my opinion, the other results in the paper (specialization to matrix completion and noisy matrix completion) are not very interesting.
Summary: The main contribution of the paper is in applying known techniques to a new problem (tensor completion), and obtaining optimal scaling in terms of the tensor dimension. However, the other results in the paper, as well as analysis are not very interesting.

Submitted by Assigned_Reviewer_7

the paper suggests a matrix reconstruction algorithm for matrices in case of incoherence in the columns only,
not necessarily in the rows. a similar result for tensors, where incoherence is in all dimensions except the
last one.


the idea is very simple. for matrices, it goes like this: recovering a column-incoherent rank-r matrice amounts
to recovering (a basis of) the column space, because given the column space you can recover each column
by sampling order of r log r entries and solving equations. to get the column space basis, you greedily
"grow" a basis by, at each iteration, trying all columns until you find one column which is not spanned
by the columns you already have. checking whether a column is not spanned can be done again by sampling O(r log r)
entries of that column. each time you find column, you add it to your basis by completely uncovering it,
hence this is an adaptive sampling strategy. it seems like a reasonable idea, although i didn't check the proof.
the same idea is applied to tensors in a recursive fashion.

compared to nuclear norm minimization techniques, the paper claims that this algorithm gives better bounds
in certain cases where the incoherence of the rows is poor (around sqrt r). the improvement is in the
logarithmic and constant terms. arguably also the algorithm here is simpler to implement (compared to SVT technology).
also the proof is simpler. I didn't check (in the appendix) whether the claim of improvement from log(n) to log(r)
is correct, but I couldn't think of a counter example for this either.

the paper also presents an algorithm and analysis for noisy matrix reconstruction, for matrices that are sum of
low rank and gaussian. the algorithm seems to be based on adaptation of Deshpande et al's approach, where
columns are sampled with probability proportional to their (sampled) projection onto the orthogonal candidate
column space. As before, in certain parameter regimes, the approach here can be better in terms of
error/sample size tradeoff.



detailed comments
p 2
"exact recover" -> "exact recovery"

"less well known" -> "less known"

p. 3
"One readily verifies that \langle x,y \rangle = vec(x)^Tvec(y) "
you didn't define \langle x,y \rangle for tensors, so there is nothing to verify. are you trying to define \langle x,y \rangle here?


p. 5
theorem 4: are you just "adapting" the proof strategy, or are you quoting the theorem from [6]?


page 6
last paragraph before section 5 - extremely hand wavy and oversimplistic.

theorem 5 is a guarantee for the algorithm described in the beginning of section 5.
why is the algorithm not written in pseudocode like algorithm 1?

p. 7
"set up" --> setup

p. 8
figure 1 - which is SVT and which is your algorithm?

It would have been nice to see a specialization algorithm 1 for matrices (which is the most interesting case).


The authors should be aware of recent work by Ailon, Chen, Huan, in which the second part is on adaptive sampling for pairwise clustering matrices (although the approach is completely different).
Summary: strength
--------
simple matrix recovery with adaptive sampling with logarithmically better bounds compared to other approaches in
certain extreme cases. also handles tensors, for which much less is known.
a recovery algorithm is also shown for the (gaussian) noise case.

weakness
--------
in other parameter regimes, the bounds of the algorithms here are worse. it would have been nice to compare
algorithms with real matrix data.
also regarding the experiments for exact recovery, it is not mentioned what the row incoherence is. it would have been
nice to see in practice how row incoherence affects the comparison between this algorithm and SVT.

Submitted by Assigned_Reviewer_8

The authors develop algorithms for performing adaptive matrix (and
tensor) completion. The authors also provide upper bounds for their
method as well as lower-bounds for the non-adaptive matrix completion
problem. I think that both results are relevant. The tensor problem
seems more interesting than the matrix version. The writing of the
paper could be improved upon.

A comparison is made to work by Gittens, where the authors of this
report state that Gittens requires row and column incoherence. To be
fair, that result is on symmetric matrices, so the statement is
vacuous. Further, the sample complexity of his result is $r n \log r$,
which is better than the $r^2$ dependency presented in this result.

The experiments do not fully explore the range of behavior of the
presented method. One regime is when the rank $r$ increases and mutual
incoherence is bounded for both the row-space and column-space. How
would the presented method behave in that setting? The theoretical
results state that there will be an $r^2$ dependency, while most
result in matrix completion establish a linear dependency on $r$.

In the computational comparison the authors state that SVT takes 2
hours. What algorithm did the authors use? For example MATLAB's built
in sparse SVD method can find the top $100$ left singular vectors of a 10k by 10k matrix in five minutes on an old laptop.

I am not 100% sure the lower-bounds are correct, but the other bounds
seem to hold.

Minor comments:

#The probability statement in Theorem 2 should be restated with
$\delta$ redefined as $\delta=\delta^'/(r^T T)$.

#The usage of $T$ changes between the noisy and noiseless setting and
$T$ is also used for transpose in the noisy section, which makes
reading the section more difficult.

#In Theorem 4 there is a typo indicating that $T$ is the probability of
sampling error.
Summary: The author's present an interesting algorithm for adaptive tensor completion and also present theoretical results to justify the results. A more thorough experimental analysis should be conducted to verify the behavior for higher order tensors.
Author Feedback

Author rebuttal: We thank the reviewers for their comments and suggestions and take this opportunity to address their most pressing concerns.

Reviewers 4, 7, and 8 all commented that more experimental results would strengthen the paper by exploring various parameter dependencies in practice. With page restrictions, we were obligated to present the experiments we found most relevant to our primary goal of verifying our theoretical results. We can add a deeper empirical study to the appendix of the paper, where we thoroughly explore our algorithms performance over a variety of parameter regimes (including on tensors).

Reviewer 7 and 8 mention that for some parameter regimes our matrix completion bound is worse than existing results. Without any assumption about ||UV^T||_{\infty} this is actually not the case as we discuss in the paragraph following Theorem 3. Moreover, such an assumption, while prevalent in the matrix completion literature, has never been established as necessary and we wanted to demonstrate what is possible without placing such assumptions on our model.

Reviewer 6 expressed concerns about the originality of the results and the justification of adaptive sampling for tensor completion. Theoretically, the insight lies in how one can use a simple primitive like estimating inner products from missing data to develop highly effective algorithms for complex tasks like matrix and tensor completion. To our knowledge, these insights have not be expressed by the machine learning community. With regards to adaptivity, while our upper bounds (as well as existing works [20]) are loose by a power of 2 in dependence on matrix/tensor rank, our bound is significantly better than all existing results for tensor completion as far as dependence on tensor size goes -- we require sample complexity proportional to the sum of the tensor dimensions where all other work (see the paragraph after Theorem 2) has dependence on the product of the tensor dimensions. Comparing with our lower bound (Theorem 4), we demonstrate an improvement by a logarithmic factor in the tensor dimensions over any algorithm that uses random sampling, which justifies investigating adaptive sampling schemes. Nevertheless, we do agree that there is a gap between our tensor completion bound and the number of parameters in the tensor and closing this gap remains an open question as we mention in the last section. It is also worth noting that tensor completion is just emerging as a problem of interest and our community's understanding is still fairly limited. We believe our results can serve as a baseline for future work on tensor completion, in particular because we establish the correct dependence on tensor dimensions (if not tensor rank).

Minor comments:
Reviewer 4 commented that we did not experimentally study the gap between our algorithm's performance and the lower bound. We can add the lower bound to the plots in Figure 1, which would demonstrate this gap.

In the plots in Figure 1 the right subspace is generated by orthogonalizing a random gaussian matrix.

In the computational comparison, we used the iterative singular value thresholding algorithm of Cai, Candes and Shen. The algorithm involves repeated computations of the SVD, which results in poor computational performance. All algorithms were implemented in python using the NumPy and SciPy libraries.

Theorem 4 is not a restatement of the theorem from [6], it is an adaptation to the tensor setting. Moreover their proof has a minor flaw which is corrected in ours.